# Emotion Regulation Difficulties as a Mediator Between Relationship Satisfaction Predicting Depressive Symptom Trajectories Among Couples in Couple Therapy

**DOI:** 10.3390/bs14121215

**Published:** 2024-12-18

**Authors:** Preston C. Morgan, Andrea K. Wittenborn, Garrin L. Morlan, Ryan Snyder

**Affiliations:** 1Human Development and Family Sciences, Oklahoma State University, Stillwater, OK 74078, USA; garrin.williams@okstate.edu (G.L.M.); ryansny@okstate.edu (R.S.); 2Human Development and Family Studies, Michigan State University, East Lansing, MI 48824, USA; andreaw@msu.edu; 3Psychiatry and Behavioral Medicine, Michigan State University, Grand Rapids, MI 49503, USA

**Keywords:** emotion regulation difficulties, depressive symptoms, relationship satisfaction, mediation, dyadic analyses

## Abstract

Although adults with depression struggle to effectively emotionally regulate themselves, these findings are limited to one partner in a romantic relationship, community samples, and cross-sectional designs. Hence, we aimed to address these gaps in the literature by investigating emotion regulation difficulties as a predictor of change in depression among couples in couple therapy. Additionally, we aimed to investigate whether emotion regulation difficulties mediated the well-established association between relationship satisfaction and changes in depression of couples in couple therapy. We examined 484 different-sex couples in couple therapy from the Marriage and Family Therapy Practice Research Network—a clinical dataset from clinics across the United States. Dyadic latent growth models revealed the actor and partner effects of emotion regulation difficulties at session 1, predicting decreases in depression trajectories across 16 sessions of couple therapy. Additionally, dyadic latent growth mediation models revealed that emotion regulation difficulties at session four did not mediate the associations between relationship satisfaction at session 1 with depression trajectories of sessions eight through 16. For couple therapy clinicians, emotion regulation difficulties can play a direct role in the treatment of depression. However, our results did not support emotion regulation difficulties as a mediator for the well-established association between relationship satisfaction and depression trajectories.

## 1. Emotion Regulation and Depressive Symptoms Among Couples in Couple Therapy

Depressive disorders are common and persistent mental health conditions [1]. Considered a “distress disorder” [2], adults with depressive disorders often experience difficulties regulating their emotions [3,4]. Findings show that emotion regulation difficulties contribute to both developing and maintaining depressive symptoms. While the relationship between emotion regulation and depression is well-established [3], the current findings on emotion regulation and depressive symptoms among couples are limited to one partner in a romantic relationship (e.g., [5]), community samples (e.g., [6]), and cross-sectional designs (e.g., [7]). Therefore, our aim was to contribute to the literature by investigating emotion regulation difficulties as a predictor of change in depression of couples in couple therapy. Furthermore, we aimed to examine whether emotion regulation difficulties mediated the association between relationship satisfaction and changes in depression among couples in couple therapy.

### 1.1. Depression Among Couples

Depressive disorders are the world’s largest contributor to global disability [1]. In the United States, approximately 21 million adults, or 8.4% of the population, have experienced at least one depressive episode in the past year [8]. Depression is increasingly recognized as a “we-disease” to reflect its profound impact on both partners [9]. Unlike some conditions that primarily affect the individual diagnosed, depressive symptoms disrupt relationship dynamics, transforming it into a shared burden. This mutual suffering is the result of a feedback loop between depressive symptoms and relationship satisfaction that exacerbates both individual and relational symptoms [10].

### 1.2. Depression and Emotion Regulation

Emotion regulation is a process of the conscious and unconscious modulation of emotions in response to environmental demands [11]. Adults with depression are more likely to use maladaptive emotion regulation strategies (e.g., rumination and suppression) and struggle to implement adaptive emotion regulation strategies (e.g., distraction and reappraisal; see [3]). Emotion regulation difficulties are both a symptom of depression and a predictor of its occurrence and persistence [12]. Individuals who experience emotion regulation difficulties are vulnerable to experiencing prolonged negative emotions, which can escalate into depressive symptoms and episodes. For example, rumination, defined as repetitive and recurring thinking about negative feelings and events, predicts the development of depressive symptoms and exacerbates existing symptoms [11]. In contrast, individuals with adaptive emotion regulation strategies (e.g., cognitive reappraisal) are more likely to experience fewer depressive symptoms. Although there is strong evidence that emotion regulation affects depression in individuals [3,12], emotion regulation difficulties can affect relationships. Partners who experience emotion regulation difficulties are more vulnerable to conflict, which perpetuates relationship distress and deteriorates mental health [13]. Additionally, how a partner emotionally regulates themselves as well as co-regulates with their partner have been shown to have effects on their own depressive symptoms [14]. To date, the empirical literature on emotion regulation and depression among couples has focused on community samples; clinical samples have not yet been examined. Although emotion regulation has been studied in relational processes and outcomes for couple therapy [15,16], it has yet to be investigated as a predictor of depressive symptoms in couple therapy.

### 1.3. Depression and Relationship Satisfaction

There is a well-established association between depression and relationship satisfaction [17]. Findings from a landmark, representative, epidemiological study demonstrated a 25-fold increase in the relative risk of depression for those in distressed relationships [18]. Relationship problems are associated with the development of future depressive symptoms [19]. The quality of an individual’s close relationships affects one’s view of themself and their sense of self-worth. Furthermore, the relationship between depression and relationship is bidirectional [20,21]. Thus, depressive symptoms put one at risk for the development of relationship distress. This association is described in the Marital Discord Model of Depression [22], a predominant theory on depression and couple relationships. It explains that improving relationship distress is associated with improvements in depressive symptoms for couples. Furthermore, this association of relationship distress and depressive symptoms extends to clinical samples, with clinical trials of couple therapy demonstrating improvements in relationship satisfaction and depressive symptoms [23]. Particularly, changes in relationship satisfaction were associated with changes in depressive symptoms for couples in a clinical trial of couple therapy for depression [24]. Despite this evidence, there continues to be a consistent call for greater understanding of the underlying mechanisms of this association [17]. Much remains unknown about *when* and *how* relationship satisfaction is associated with depressive symptoms. We aim to heed this call by examining emotion regulation difficulties as a mediator. 

### 1.4. Emotion Regulation as a Mediator

There is support for emotion regulation as a mediator between relationship status and depression [5,25] as well as insecure attachment and depression [6]. However, emotion regulation has not been tested as a mediator for relationship satisfaction and depressive symptoms. Put simply, it remains unknown whether emotion regulation explains *how* relationship satisfaction is associated with depressive symptoms. One aspect of the Marital Discord Model of Depression is the acceptance of emotions in the romantic relationship, which encourages partners to disclose and be forthcoming with each other [22]. Conversely, partners may have difficulty with emotionally regulating themselves when there is little to no acceptance of negative emotions. The Marital Discord Model of Depression suggests that improving relationship satisfaction would *then* create more acceptance of emotions, which would improve emotion regulation and *then* improve depressive symptoms. Thus, we expect emotion regulation difficulties to mediate relationship satisfaction and depressive symptoms. 

Prior investigations of emotion regulation as a mediator were conducted with cross-sectional designs, sampled only one partner, and/or examined non-clinical samples. For example, one study had a large sample of 630 adults who participated in an online survey at one time [6]. Another studied two large samples (1319 adults and a national sample of 772 adults) with romantic relationships recruited through the random dialing of telephone numbers and an online sample at one time point [5]. This highlights the need for longitudinal studies, clinical samples, and the inclusion of both partners. 

### 1.5. Current Study

Our aim was to investigate emotion regulation among couples with depressive symptoms over the course of couple therapy. This study was designed in response to several important gaps in the literature on emotion regulation and depression among couples. First, most of the literature sampled only individual adults or one partner of romantic relationships, which limits emotion regulation to the perception of one person in the relationship. Second, the literature primarily sampled community or population samples, so these results may not apply to clinical samples. Third, despite emotion regulation being tested as a mediator, the studies were most commonly cross-sectional, severely limiting our understanding of the mediating role of emotion regulation over time. Using clinical data from the Marriage and Family Therapy Practice Research Network, a dataset from marriage and family therapy clinics across the U.S., we aimed to test the following research questions: 

*RQ1*: To what extent are both partners’ emotion regulation difficulties associated with their own and their partner’s depressive symptoms over the course of couple therapy?

*RQ2*: To what extent do both partners’ emotion regulation difficulties mediate the associations between their own and their partner’s relationship satisfaction and depressive symptoms?

## 2. Methods

We used data from the Marriage and Family Therapy Practice Research Network (MFT-PRN) [26], which includes clinical data from marriage and family therapy university clinics and community settings across the U.S. All clients consented to the MFT-PRN and were provided the same treatment services they would have received if they declined to consent. Prior to each session, clients completed electronic surveys that were stored in an electronic database: MFT-PRN. Specifically, we used data from couples in couple therapy from sessions 1, 4, 8, 12, and 16. Given the clinical nature of the data, this dataset was not publicly available [26]. The lead author’s university deemed the data for this study to be exempt. We limited the data to couples in which both partners completed the emotion regulation, depression, and relationship satisfaction measures at session 1 (*n* = 517 couples). There was 1 same-sex couple and 32 with missing data on sex that were removed from the sample, which resulted in a final sample of 484 different-sex couples. 

### 2.1. Sample Characteristics

Sample characteristics were completed at the first or second session. These couples were in committed relationships with 22% not currently living together and the remaining 78% currently living together. On average, couples were together for 4.52 years (*SD* = 7.2, *Median* = 2, *Range*, 0–48) and had an average income in the category of USD 30,000–$39,999. On average, partners were in their late 20s (Men: *M* = 28.08, *SD* = 8.8, *Median* = 25, *Range* = 19–72; Women: *M* = 27.08, *SD* = 8.37, *Median* = 24, *Range* = 18–70). Men identified as White/Caucasian (83%), Hispanic/Latino/Spanish (6%), Black/African American (0.02%), Asian (1.6%), American Indian/Alaska Native (0.06%), Native Hawaiian/Pacific Islander (0.08%), and multiracial or endorsed more than one race (6.6%). Women identified as White/Caucasian (81%), Hispanic/Latino/Spanish (8%), Asian (2%), Native Hawaiian/Pacific Islander (0.02%), and multiracial or endorsed more than one race (7.8%). Some partners had a bachelor’s degree (4-year college degree) or higher (Men: 38%; Women: 45%). On average, the partners reported no pressure to attend couple therapy (Men: *M =* 1.77; Women: *M* = 1.28), while many had previous experience with therapy (Men: 64%; Women: 76%) and some were in concurrent individual therapy (Men: 13%; Women: 21%). Some partners reported taking anti-depressant medication (Men: 10%; Women: 18%). Although dates at session 1 were not provided, the length of treatment from session 2 to session 16 was an average of 22.58 weeks (*SD* = 6.54, *Median* = 21.29, *Range*, 10.14–16.57). Finally, partners reported an average treatment progress where the problems were a little better (Men: *M* = 5.17, *SD* = 0.85, *Median* = 5.22, *Range* = 1.67–7; Women: *M* = 5.03, *SD* = 0.86, *Median* = 5, *Range* = 1–7)

### 2.2. Measures

#### 2.2.1. Emotion Regulation

Difficulties with emotion regulation was measured by the Difficulties in Emotion Regulation Scale Short Form (DERS-SF) [27], which is a valid, reliable, and briefer version of the full Difficulties in Emotion Regulation Scale. This 18-item scale ranges from “I have no idea how I am feeling” to “When I’m upset, I lose control over my behavior” and is rated from 1 (*Almost never*) to 5 (*Almost always*). These items were summed together to create a variable where higher scores indicated higher difficulties in emotion regulation. The DERS-SF had an acceptable internal reliability (i.e., Cronbach’s alpha) for men (session 1: *α =* 0.87, session 4: *α =* 0.86) and women (session 1: *α =* 0.87, session 4: *α =* 0.88). 

#### 2.2.2. Depression

Depression was measured by the 9-item Patient Health Questionnaire (PHQ-9) [28]. The PHQ-9 is a reliable and valid measure of depression that is sensitive to changes in depression over time [28]. The nine items ranged from “little interest or pleasure in doing things” to “thoughts that you would be better off dead or of hurting yourself in some way”, which were rated from 0 (*Not at all*) to 3 (*All the time*). Items were summed to create a variable where higher scores indicated higher levels of depression. Scores above 10 met the clinical cut-off for depression. This had an acceptable internal reliability for men (Session 1: *α =* 0.86, session 4: *α =* 0.84, session 8: *α =* 0.85, session 12: *α =* 0.86, session 16: *α =* 0.85) and women (session 1: *α =* 0.87, sessions 4, 8, 12, and 16: *α =* 0.84). 

#### 2.2.3. Relationship Satisfaction

Relationship satisfaction was measured by the 16-item Couple Satisfaction Index (CSI) [29] that assessed partners’ overall levels of satisfaction in their romantic relationship. The items ranged from “In general, how often do you think that things between you and your partner are going well” to “I really feel like part of a team with my partner”. These items were summed together to create a variable where higher scores indicated relationship satisfaction. Scores below 51.5 indicate relationship distress. This had an acceptable internal reliability for men at session 1 (*α =* 0.98) and women at session 1 (*α =* 0.98).

#### 2.2.4. Controls

We included client characteristics and treatment factors as controls including relationship duration (*Years*), race and ethnicity (*White = 1*, *Black*, *Asian*, *American Indian*, *Middle Eastern*, *and multiracial = 0*), income (*11 categories from < USD10*,*000 to greater than $1*,*000*,*000*), sexual orientation (G*ay*, *lesbian*, *or bisexual =* 1, *heterosexual or straight* = 0), pressure to attend therapy (*Single item from 1 to 5 with higher scores indicating more pressure to attend therapy)*, in current treatment with an individual therapist *(concurrent treatment = 1*, *no concurrent treatment = 0*), and antidepressant medication (*antidepressant medication =* 1, *no antidepressant medication* = *0*). Treatment progress was measured by an average of three items that asked clients to rate the progress of the top three biggest problems on a scale of 1 (*Problem is much worse*) to 7 (*Problem is solved/Much better*). These items were given after session 1 and averaged together to create an average treatment progress across the course of treatment. Anxious and avoidant attachment were assessed using the Experiences in Close Relationships Scale-Short Form [30], which had an acceptable Cronbach’s alpha for anxious attachment (men: *α =* 0.72; women: *α =* 0.71) and avoidant attachment (men: *α =* 0.81; women: *α =* 0.79). Both partners’ scores on income, relationship duration, and length of treatment were averaged together to create a single variable for each of the following: income, relationship duration, and length of treatment. 

### 2.3. Analytic Plan 

We used R [31] to run descriptive and bivariate correlations of the key variables. Several variables had high skewness and kurtosis (i.e., relationship duration, length of treatment between sessions 2 and 4 as well as 4 and 8, and both partners’ pressure to attend therapy), and were logarithmic transformed. Next, the following analyses were completed using Mplus [32]. Dyadic latent growth curve modeling [33] was used to test research question one. For the unconditional model, the women’s initial and rates of changes were latent constructs comprised of women’s observed depression at sessions 1, 4, 8, 12, and 16, which had factor loadings of 0, 1, 2, 3, and 4, respectively. This same procedure was followed for men’s depressive symptom trajectories. Both men’s and women’s error variances at each time point were correlated. We evaluated possible nonlinear trends by freely estimating parameters as well as adding quadratic and cubic latent constructs. With good model fit, we then developed a conditional model by adding difficulties in emotion regulation and controls as predictors of men’s and women’s initial and rates of changes in depressive symptom trajectories. 

Dyadic mediation and latent growth curve modeling was used to test research question two. For the unconditional model, men’s and women’s initial and rates of change in depressive trajectories were from sessions 8, 12, and 16 with factor loadings of 2, 3, and 4, respectively. With good model fit, we then created the conditional model of men’s and women’s relationship satisfaction, along with controls, at session 1, predicting the men and women’s emotion regulation difficulties at session 4, which in turn predicted men’s and women’s depressive trajectories from sessions 8, 12, and 16 (see Figure 1). Of the controls, men’s and women’s treatment progress predicted men’s and women’s initial and rates of change in depressive symptoms. Additionally, the length of treatment between sessions 2 and 4 predicted both partners’ emotion regulation difficulties at session 4, and the length of treatment between sessions 4 and 8 predicted both partners’ initial and rates of change in depressive trajectories. Mediation was analyzed by grand mean centering all continuous variables, binary variables coded as 1, −1, and by using model indirect effects with 2000 bootstraps. Good model fit was evaluated by a comparative fit index (CFI) greater than 0.95 as well as a standardized root mean square residual (SRMR) and root mean square error of approximation (RMSEA) less than 0.05 [34].

*Notes:* Both partners’ treatment progress and length of treatment (sessions 4–8) were the only controls to predict men’s and women’s initial and rates of changes in depressive symptom trajectories. Square boxes represent observed variables. Circles represent latent constructs. Men’s and women’s initial and rates of changes in depressive symptom trajectories are a dyadic latent growth model comprised of observed depressive symptoms at sessions 8, 12, and 16. 

#### Missing Data

Of the variables at session 1, missingness ranged from 0.41% (e.g., men’s medication) to 26.70% (e.g., men’s average treatment progress). As to be expected with clinical data, there was a noticeable amount of missingness for depression at session 4 (28.3% women, 31.2% men), session 8 (49.4% women, 51.7% men), session 12 (65.5% women, 68.8% men), and session 16 (75.4% women, 78.9% men). There was also attrition for missingness on emotion regulation difficulties at session 4 (31.00% women, 28.50% men). Little’s tests (from the *naniar* package; [35]) revealed that missingness was completely missing at random for the RQ1 model (*χ*^2^ (1851) = 1936, *p* = 0.08) and RQ2 model (*χ*^2^ (1690) = 1781, *p* = 0.06). To further evaluate the missingness, we conducted Pearson correlation tests of the key variables. We coded missingness (*Missingness =* 1, *Observed* = 0) of depression and emotion regulation difficulties at sessions 4, 8, 12, and 16. Results revealed that missingness on depression at sessions 4, 8, 12, and 16 and emotion regulation difficulties at session 4 were associated with at least two variables. For example, men’s avoidant attachment was associated with their own and their partner’s lower missingness at sessions 8, 12, and 16 (*r* = −0.13 was the lowest and *r* = −0.19 was the highest). Given these missingness analyses, we used full-information maximum likelihood to handle missingness in our models. 

## 3. Results

The preliminary correlation analyses are reported in Table 1 and summarized here. Correlations revealed that emotion regulation difficulties at session 1 was positively associated with depressive symptoms at each session for both partners (See Table 1). Specifically, women’s higher emotion regulation difficulties at session 1 was consistently associated with their own and their partner’s higher depressive symptoms at each later session. However, men’s higher emotion regulation difficulties at session 1 were consistently associated with their own higher depressive symptoms at each session, but only associated with their partner’s higher depressive symptoms at sessions 1 and 4. Interestingly, men’s and women’s higher relationship satisfaction were only associated with men’s lower emotion regulation difficulties at session 4, but not with women’s emotion regulation at session 4. This may suggest a weak association between relationship satisfaction and emotion regulation difficulties. 

Next, we began to test research question one with an unconditional dyadic latent growth model of men’s and women’s depressive symptom trajectories of sessions 1, 4, 8, 12, and 16, and then evaluated the nonlinear trends in the model. These were non-positive definite in the latent covariance matrix when we accounted for quadratic, cubic, and freely estimating slope parameters, but not in the linear slope model. Hence, we used the linear slope model. Results reported here had a *p* < 0.05 unless otherwise specified; refer to the tables for details. The linear unconditional dyadic latent growth model had an acceptable model fit: CFI = 0.98, RMSEA = 0.04 (95% CI 0.2–0.6), and SRMR = 0.04. Men started with mild initial levels of depressive symptoms at session 1 (*b* = 6.40) that varied across couples (*σ*^2^ = 18.18), which then decreased (*b =* 0.17) over the course of couple therapy and did not vary across couples (*σ*^2^ = 0.17, *p* = 0.33). Women also started with mild initial levels of depressive symptoms at session 1 (*b* = 7.48) that varied across couples (*σ*^2^ = 21.42), which then did not change over the course of couple therapy (*b* = −0.16, *p* = 0.06), but varied across couples (*σ*^2^ = 0.62). Particularly, women started with higher depressive symptoms than men (*χ*^2^ (1) = 14.69, *p* < 0.01.), but the linear rates of change were not different between men and women (*χ*^2^ (1) = 0.28, *p* = 0.65).

Next, we added predictors and control variables to the model. This resulted in a model with a non-positive definite covariance matrix, which was resolved by fixing the variance of women’s linear rates of change to 0. This conditional model also had an acceptable model fit: CFI = 0.95, RMSEA = 0.04 (95% CI 0.3 to 0.4), and SRMR = 0.03. Men’s higher emotion regulation difficulties at session 1 was associated with their own (*b* = 1.94) and their partner’s higher levels of initial depressive symptoms (*b* = 0.60). Women’s higher emotion regulation difficulties at session 1 was associated with their own (*b* = 2.14) and their partner’s higher levels of initial depressive symptoms (*b* = 0.59) as well as decreasing rates of change in their own (*b* = −0.18) and their partner’s depressive symptom trajectories (*b* = −0.24). See Table 2 for further details. 

Finally, to test research question two, we began with an unconditional dyadic latent growth model of the men and women’s depressive symptom trajectories from sessions 8, 12, and 16. This model revealed a non-positive definite covariance matrix, which was resolved when we fixed the variances of both men’s and women’s rates of change to 0, which then had an acceptable model fit: CFI = 1.00, RMSEA = 0.0 (95% CI 0 to 0.4), and SRMR = 0.05. Men started with mild initial levels of depressive symptoms at session 8 (*b* = 6.39) that varied across couples (*σ*^2^ = 16.69), which did not change over the course of couple therapy (*b* = −0.07, *p* = 0.69) and did not vary across couples (fixed to 0). Women also started with mild initial levels of depressive symptoms at session 8 (*b* = 8.11) that varied across couples (*σ*^2^ = 16.35), which then did not change over the course of couple therapy (*b* = −0.18, *p* = 0.32) and did not vary across couples (fixed to 0). Particularly, women started with higher depressive symptoms than men (*χ*^2^ (1) = 14.16, *p* < 0.01) but the linear rates of change were not different between men and women (*χ*^2^ (1) = 0.24, *p* = 0.62).

Next, we added the predictors and controls to create the dyadic latent growth mediation model. This model had an acceptable model fit: CFI = 0.97, RMSEA = 0.02 (95% CI 0 to 0.3), and SRMR = 0.04. With an acceptable model fit, we then tested indirect pathways from men’s and women’s relationship satisfaction at session 1 to men’s and women’s initial and rates of change in depressive symptoms. Men’s and women’s higher relationship satisfaction at session 1 was neither associated with their own nor their partner’s higher emotion regulation difficulties at session 4. However, men’s higher emotion regulation difficulties at session 4 was associated with their own higher initial levels of depressive symptoms at session 8 (*b* = 0.26) while controlling for men’s and women’s treatment progress and relationship satisfaction as well as length of treatment from sessions 4 to 8. Women’s higher emotion regulation difficulties at session 4 was associated with their own higher initial levels of depressive symptoms at session 8 (*b* = 0.22). See Table 3 for details. We then tested the indirect effects using the bootstrap method (i.e., 2000 bootstraps) for confidence intervals. These results revealed no indirect effects from relationship satisfaction to emotion regulation difficulties to the initial levels and rates of change in depressive symptom trajectories for actor or partner effects (See Table 4). Put simply, men’s and women’s emotion regulation at session 4 did not mediate the associations between men’s and women’s relationship satisfaction at session 1 with their own and their partner’s initial levels and rates of change in depressive symptom trajectories from sessions 8 to 16.

## 4. Discussion

We aimed to examine the associations between both partners’ emotion regulation difficulties with their own and their partner’s depressive symptoms over the course of couple therapy. We also tested whether both partners’ emotion regulation difficulties mediated associations between their own and their partner’s relationship satisfaction and depression symptoms. Results revealed that emotion regulation difficulties at intake were associated with the initial and rates of change in depressive trajectories over the course of couple therapy (RQ1). Furthermore, we examined the potential mechanism of emotion regulation difficulties for the association between relationship satisfaction and depressive trajectories. The results revealed that emotion regulation difficulties was not a mediator for this association (RQ2). These findings contribute to the literature in two ways. First, the findings expand on the support of emotion regulation difficulties predicting depressive symptoms from individuals to couples in couple therapy. The previous literature focused on emotion regulation and depression in individuals in community samples or clinical samples [3]. By focusing on couples, these findings demonstrated actor and partner effects, which are discussed below. Second, we heeded a consistent call to explore the mechanisms for the well-established association between depression and relationship dissatisfaction [3]. Our findings greatly contribute to this literature by examining this association in a clinical sample as well as the mediation effects. These results provide new findings for researchers to replicate and examine in future work. Taken together, we discuss two important findings. 

### 4.1. Emotion Regulation Difficulties as a Predictor of Depressive Symptoms

For both partners, more emotion regulation difficulties were associated with higher levels of depressive symptoms at intake, for themselves and their partners. The actor effects were consistent with the previous literature; both men and women who have difficulty regulating negative emotions tend to experience more frequent depressive symptoms [3,36,37]. However, the partner effects contribute to the literature to show that even higher emotion regulation difficulties are associated with their partner’s higher levels of depressive symptoms. This highlights the interdependent nature in close relationships and underscores the importance of considering both individual and dyadic processes in couple therapy, as each partner’s emotions impact not only their own mental health, but also their partner’s. This has implications for clinicians to not only expect couples to present at intake with higher emotion regulation difficulties, but that there may be an interaction effect between partners. Particularly, men’s and women’s higher depressive symptoms at intake may be a part of their own emotion regulation difficulties *and* their partner’s emotion regulation difficulties. 

Next, our findings revealed that emotion regulation was not only a predictor of the initial levels of depressive symptoms, but decreasing rates of change in depressive symptom trajectories. Specifically, women’s higher emotion regulation difficulties were associated with decreases in their own and their partner’s trajectories of depressive symptoms across 16 sessions of couple therapy. One possible explanation for this is that difficulties in emotion regulation may surface due to unresolved relational issues during therapy, making them more visible and addressable [38]. Couple therapy often targets emotion regulation by promoting healthier emotional communication and coping strategies for both partners [28,39]. Given that women tend to be more emotionally expressive in relationships [40], their struggles with emotion regulation might create more opportunities for therapeutic engagement and progress. As women’s difficulties in emotion regulation are addressed in therapy, the overall relational dynamic may improve, benefiting both partners’ emotional well-being. However, due to the naturalistic setting of this study, we cannot definitively conclude that emotion regulation was directly targeted or treated. Future research in more controlled settings is needed to clarify the mechanisms by which women’s emotion regulation difficulties influence depressive symptom changes in couple therapy. 

Relatedly, another possible explanation for this may be that women often take the lead in seeking couple therapy [41] and display greater willingness to engage in mental health care [42,43]. They also report more effort toward self-improvement in therapy compared to men [44]. This higher level of engagement may explain why women’s satisfaction is closely tied to the couple’s progress in therapy, as their satisfaction likely reflects a more effective therapeutic outcome that benefits both partners’ depressive symptoms. 

Additionally, another explanation could be gender differences and similarities with emotion regulation and psychopathology [27,45]. For example, compared to men, women use different emotion regulation strategies [46], but there are gender similarities in that emotion regulation is associated with psychopathology (e.g., depression; [45]). In the current study, both genders had similar moderate averages of emotion regulation difficulties at session 1 (Men: *M* = 40.56; Women: *M* = 41.71) and session 4 (Men: *M* = 40.43; Women: *M* = 40.78). However, our finding that women’s emotion regulation plays a role in both partners’ depressive symptoms is inconsistent with the noted prior literature. Prior studies found gender differences in specific strategies. For instance, higher rumination is related to more depression and anxiety for women compared to men [47]. A recent study offers a more detailed look at the underlying factors behind gender differences in emotion processing. Using an evidence accumulation model to analyze reports of emotions, the study found that women are able to generate negative emotions more quickly than men [48]. This suggests that women might be more sensitive to negative emotions, potentially linked to their evolutionary role as primary caregivers, which would require a heightened awareness of threats to their children’s safety [48]. This increased emotional sensitivity could help explain why women, even though they use more emotion regulation strategies, are more prone to depression and anxiety. Future research could explore how gender and emotion regulation strategies interact, particularly in relation to mental health outcomes. Based on these findings, clinicians can address difficulties with emotion regulation through the implementation of targeted interventions such as emotion regulation skills and coping strategies. Interestingly, our findings highlight that women’s emotion regulation difficulties play a key role in decreasing their own and their partner’s depressive symptoms among couples in couple therapy.

### 4.2. Emotion Regulation Difficulties Did Not Mediate

The finding that emotion regulation difficulties did not mediate the well-established association between relationship dissatisfaction and depression was surprising. Based on the Marital Discord Model of Depression, we expected to find that improving relationship satisfaction would *then* improve emotion regulation, which would *then* improve depressive symptoms. However, our findings do not support this mediation. One reason for the lack of mediation may be attributed to the length of time between time points. Although we accounted for this in the model, emotion regulation difficulties may show up prior to session 4 in couple therapy. Another major reason may be due to the weak association (found in the correlation results) or lack of association (found in the conditional models) between both partners’ relationship satisfaction at session 1 with their own and their partner’s emotion regulation difficulties at session 4. It is possible that this could be due to relationship satisfaction being an *interpersonal* construct and emotion regulation being an *intrapersonal* construct. For example, these findings suggest that improving one’s satisfaction with their romantic relationship does not affect how they emotionally regulate themselves. Rather, this may be due to the sample being a clinical sample where the couples are in relationship distress and struggling to regulate their emotions. It is possible that emotion regulation difficulties may *affect* how satisfied a partner is with their relationship, which would suggest that the direction of this relationship may be switched. Previously, in testing theoretical concepts from Coyne’s Interpersonal Theory of Depression [49], another major theory on couples with depression, the association between emotion regulation difficulties and depressive symptoms was mediated by excessive reassurance seeking (i.e., a key theoretical concept in Coyne’s theory; [50]. In this case, emotion regulation was not a mediator but a predictor. Hence, future research should examine whether relationship satisfaction is a mediator between emotion regulation difficulties and depression. Together, the clinical implications of this may suggest a different temporal association. Although it is clear that emotion regulation difficulties play a role in couple therapy, *how* they play a role in couple therapy remains unclear. 

### 4.3. Limitations

There are both strengths and limitations associated with using clinical data from naturalistic settings in this study. Particularly, the findings apply to real-world practices where clients present for couple therapy. Additionally, this dataset enabled us to analyze couples both dyadically and longitudinally. However, the naturalistic setting limited our ability to identify and control other potential influences on the couple outcomes. Notably, we lacked information on treatment factors such as presenting problems, interventions, and clinic procedures utilized. Another limitation was the attrition rate in our sample. Our examination of missing data showed a reduction in the number of participants attending later sessions, which is a common challenge in clinical research. This may be attributed to various external factors including financial constraints, significant life changes, or dissatisfaction with treatment, potentially biasing our results toward couples who had resources or were motivated to complete more therapy [51]. Finally, the findings of this study may not be fully generalizable to more diverse populations, as the sample predominantly consisted of White, middle-class couples. Despite these limitations, this study provides valuable insights into links between emotion regulation, relationship satisfaction, and depression in couple therapy.

## 5. Conclusions

We found that emotion regulation was a predictor of depressive symptom trajectories for both partners. Particularly, these results demonstrate that emotion regulation may play an important role in treating couples with depressive symptoms. However, we also found that emotion regulation did not mediate between relationship satisfaction and depressive symptom trajectories. This surprising finding demonstrates that the order of improving relationship satisfaction, then emotion regulation to decrease depressive symptoms may need to be re-considered. It may be that emotion regulation should be an earlier focus (before session 4) in couple therapy. These findings are from clinical data from naturalistic settings, meaning that couples presented for a variety of reasons and received a variety of couple therapy interventions. In real-world clinical settings, emotion regulation appears to be an important role in treating depressive symptoms for couples and not just individuals. 

## Figures and Tables

**Figure 1 behavsci-14-01215-f001:**
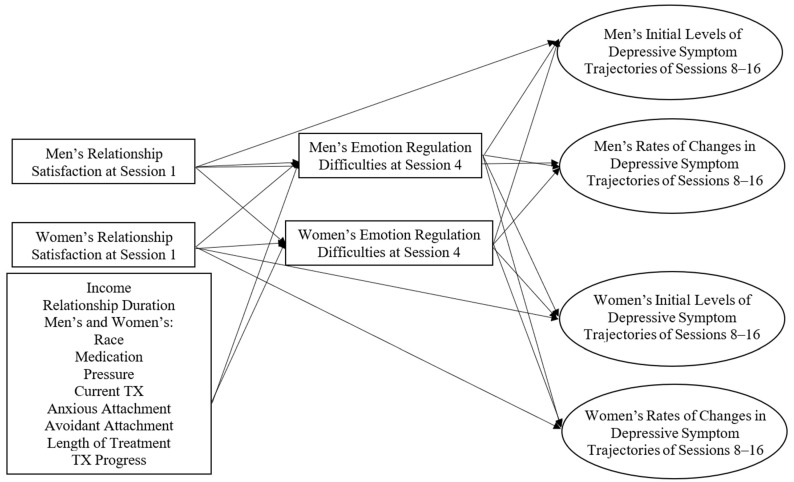
Conceptual Model of the Dyadic Latent Growth Mediation Model.

**Table 1 behavsci-14-01215-t001:** Correlations of the predictors and outcome variables.

	Men’s Outcome Variables						Women’s Outcome Variables					
	DER S4	DS S1	DS S4	DS S8	DS S12	DS S16	DER S4	DS S1	DS S4	DS S8	DS S12	DS S16
Men’s RS	**−0.18**	**−0.40**	**−0.30**	**−0.26**	**−0.27**	−0.16	−0.09	**−0.25**	**−0.14**	−0.11	0.02	0.00
Women’s RS	**−0.12**	**−0.29**	**−0.18**	**−0.14**	−0.13	−0.04	−0.10	**−0.30**	**−0.22**	−0.11	−0.04	0.05
Men’s DER	**0.74**	**0.54**	**0.42**	**0.45**	**0.52**	**0.40**	0.08	**0.16**	**0.13**	**0.21**	**0.19**	**0.21**
Women’s DER	0.01	**0.14**	**0.15**	0.06	0.12	0.16	**0.76**	**0.54**	**0.48**	**0.48**	**0.31**	**0.45**
Relationship Duration	−0.02	0.02	−0.03	−0.04	−0.01	−0.06	−0.10	−0.01	−0.04	−0.11	−0.15	−0.15
Income	0.00	−0.05	−0.07	−0.11	−0.10	−0.11	−0.08	−0.03	**−0.16**	**−0.18**	**−0.21**	−0.17
Men’s Race	**0.12**	0.00	0.09	**0.14**	0.12	0.13	0.01	−0.01	0.04	0.01	0.16	0.05
Women’s Race	0.00	0.03	0.07	0.07	0.04	0.00	−0.05	−0.01	0.02	0.02	0.14	0.07
Men’s Medication	0.05	**0.14**	**0.15**	0.09	**0.12**	0.07	0.02	−0.01	0.01	0.06	0.00	−0.12
Women’s Medication	0.04	**0.17**	**0.15**	**0.13**	**0.21**	0.17	0.08	**0.21**	**0.15**	**0.16**	**0.23**	0.17
Men’s Avoidant Attachment	**0.24**	**0.38**	**0.22**	**0.23**	**0.22**	0.11	0.01	**0.14**	0.09	0.02	−0.08	−0.03
Women’s Avoidant Attachment	0.07	**0.20**	0.07	0.07	0.07	−0.02	**0.16**	**0.27**	**0.25**	**0.18**	0.03	−0.03
Men’s Anxious Attachment	**0.21**	**0.32**	**0.29**	**0.31**	0.31	0.16	0.02	**0.13**	**0.13**	0.06	0.01	0.03
Women’s Anxious attachment	0.02	0.06	−0.01	−0.01	0.03	0.09	**0.37**	**0.38**	**0.33**	**0.21**	**0.21**	**0.25**
Men’s Pressure	0.03	**0.12**	0.04	−0.03	0.02	0.12	−0.02	0.02	0.01	0.05	0.07	0.07
Women’s Pressure	−0.06	0.07	0.04	**0.19**	0.00	0.13	**0.16**	**0.15**	0.04	**0.13**	**0.16**	0.05
Men’s Current TX	0.06	**0.12**	**0.21**	0.06	0.06	0.06	−0.01	0.00	−0.01	−0.09	−0.03	−0.03
Women’s Current TX	−0.05	0.05	0.06	0.02	0.01	0.01	0.07	**0.14**	0.10	0.06	−0.01	−0.02
Men’s Treatment Progress	**−0.16**	**−0.22**	**−0.29**	**−0.31**	**−0.36**	**−0.34**	−0.03	**−0.07**	−0.11	−0.09	**−0.08**	−0.19
Women’s Treatment Progress	**−0.13**	**−0.15**	**−0.15**	**−0.18**	**−0.21**	**−0.36**	**−0.13**	**−0.20**	**−0.24**	**−0.35**	**−0.37**	**−0.20**

*Notes:* DER = emotion regulation difficulties. DS = depressive symptoms. S1 = session 1, S4 = session 4, S8 = session 8, S12 = session 12, S16 = session 16. TX = treatment. ***Bolded r***, *p <* 0.0*5.*

**Table 2 behavsci-14-01215-t002:** Conditional dyadic latent growth curve model of men’s and women’s depressive symptom trajectories (N = 484).

	Men’s Initial Depressive Symptoms				Men’s Rates of Changes in Depressive Symptoms				Women’s Initial Depressive Symptoms				Women’s Rates of Changes in Depressive Symptoms			
	*b*	*SE*	*p*	*β*	*b*	*SE*	*p*	*β*	*b*	*SE*	*p*	*β*	*b*	*SE*	*p*	*β*
Men’s DER	**1.94**	0.18	0.00	0.48	−0.11	0.07	0.12	−0.23	**0.60**	0.18	0.00	0.14	0.03	0.07	0.64	0.06
Women’s DER	**0.59**	0.19	0.00	0.15	**−0.18**	0.08	0.03	−0.36	**2.14**	0.19	0.00	0.50	**−0.24**	0.08	0.00	−0.41
Men’s RS	**−0.50**	0.22	0.02	−0.18	0.09	0.09	0.34	0.26	0.16	0.22	0.45	0.06	−0.08	0.09	0.38	−0.20
Women’s RS	−0.24	0.18	0.19	−0.09	0.09	0.07	0.21	0.29	**−0.74**	0.18	0.00	−0.28	**0.14**	0.07	0.05	0.40
Relationship Duration	0.13	0.24	0.60	0.03	0.03	0.10	0.73	0.06	0.06	0.24	0.82	0.01	−0.12	0.10	0.21	−0.19
Income	−0.12	0.07	0.08	−0.09	0.00	0.03	0.96	−0.01	**−0.15**	0.07	0.03	−0.10	0.00	0.03	0.89	−0.02
Men’s Race	−0.42	0.53	0.43	−0.04	0.00	0.21	0.99	0.00	−0.08	0.53	0.88	−0.01	0.12	0.21	0.58	0.07
Women’s Race	**1.14**	0.50	0.02	0.10	−0.17	0.20	0.38	−0.13	0.64	0.51	0.21	0.06	−0.08	0.21	0.68	−0.05
Men’s Medication	1.06	0.62	0.09	0.07	−0.29	0.25	0.24	−0.17	−0.10	0.62	0.87	−0.01	−0.20	0.25	0.42	−0.10
Women’s Medication	0.69	0.49	0.16	0.06	−0.05	0.19	0.79	−0.04	**1.50**	0.49	0.00	0.13	−0.11	0.19	0.58	−0.07
Men’s Pressure	0.47	0.37	0.21	0.05	−0.08	0.16	0.62	−0.07	**−1.05**	0.37	0.01	−0.11	**0.38**	0.16	0.02	0.30
Women’s Pressure	0.14	0.52	0.79	0.01	0.08	0.21	0.69	0.06	−0.03	0.53	0.96	0.00	0.09	0.21	0.68	0.05
Men’s Current TX	0.63	0.60	0.29	0.05	−0.12	0.27	0.66	−0.07	−0.78	0.60	0.19	−0.06	0.15	0.28	0.58	0.08
Women’s Current TX	0.15	0.49	0.76	0.01	−0.01	0.20	0.96	−0.01	0.71	0.49	0.15	0.06	**−0.45**	0.20	0.03	−0.29
Men’s Avoidant Attachment	0.06	0.04	0.19	0.08	−0.02	0.02	0.17	−0.29	−0.07	0.04	0.13	−0.09	0.00	0.02	0.97	0.01
Women’s Avoidant Attachment	−0.07	0.04	0.06	−0.11	0.02	0.02	0.36	0.18	0.00	0.04	0.97	0.00	−0.02	0.02	0.33	−0.16
Men’s Anxious Attachment	**0.07**	0.03	0.03	0.11	0.00	0.01	0.75	−0.05	0.02	0.03	0.60	0.03	−0.01	0.01	0.44	−0.11
Women’s Anxious Attachment	−0.04	0.03	0.19	−0.07	0.02	0.01	0.24	0.20	**0.13**	0.03	0.00	0.20	**−0.03**	0.01	0.03	−0.32
Men’s TX Progress	**−0.61**	0.28	0.03	−0.12	−0.20	0.12	0.08	−0.33	−0.04	0.27	0.87	−0.01	−0.02	0.12	0.89	−0.02
Women’s TX Progress	0.13	0.27	0.62	0.03	**−0.23**	0.11	0.04	−0.37	**−0.78**	0.26	0.00	−0.15	**−0.34**	0.11	0.00	−0.48

*Notes:* DER = emotion regulation difficulties. DS = depressive symptoms. S1 = session 1, S4 = session 4, S8 = session 8, S12 = session 12, S16 = session 16. TX = treatment. ***Bolded b***, *p <* 0.05.

**Table 3 behavsci-14-01215-t003:** Conditional dyadic mediation latent growth model for both partners’ depressive symptom trajectories (*n* = 484).

	Men’s Initial Levels				Men’s Rates of Changes				Women’s Initial Levels				Women’s Rates of Changes			
	*b*	*SE*	*p*	*β*	*b*	*SE*	*p*	*β*	*b*	*SE*	*p*	*β*	*b*	*SE*	*p*	*β*
Men’s DER (S4)	**0.26**	0.05	0.00	0.56	−0.02	0.02	0.17	−0.44	0.08	0.05	0.15	0.17	−0.01	0.02	0.55	−0.26
Women’s DER (S4)	0.04	0.05	0.49	0.08	0.00	0.02	0.89	−0.05	**0.22**	0.05	0.00	0.51	−0.02	0.02	0.39	−0.40
Men’s RS	−0.07	0.05	0.13	−0.24	0.01	0.02	0.49	0.33	0.01	0.05	0.84	0.03	0.00	0.02	0.96	−0.03
Women’s RS	0.00	0.04	0.94	−0.01	0.00	0.02	0.87	0.08	−0.01	0.04	0.77	−0.05	0.01	0.02	0.74	0.19
Men’s TX Progress ^a^	**−1.93**	0.77	0.01	−0.35	0.19	0.26	0.48	0.29	1.46	0.79	0.07	0.27	−0.48	0.28	0.09	−0.96
Women’s TX Progress ^a^	1.43	0.78	0.07	0.26	**−0.60**	0.27	0.03	−0.96	**−2.26**	0.77	0.00	−0.42	0.15	0.27	0.57	0.31
Treatment Length (4–8)	−0.44	1.45	0.76	−0.04	0.07	0.52	0.89	0.06	−0.69	1.38	0.62	−0.06	0.09	0.50	0.86	0.09
					Men’s DER at Session 4				Women’s DER at Session 4							
					*b*	*SE*	*p*	*β*	*b*	*SE*	*p*	*β*				
Men’s RS					0.03	0.06	0.66	0.04	−0.04	0.06	0.56	−0.05				
Women’s RS					0.02	0.05	0.78	0.03	0.01	0.05	0.92	0.01				
Relationship Duration					−0.85	0.69	0.22	−0.08	0.11	0.69	0.87	0.01				
Income					0.08	0.20	0.69	0.02	−0.23	0.19	0.24	−0.07				
Men’s Race ^b^					**1.98**	0.82	0.02	0.14	0.81	0.82	0.32	0.06				
Women’s Race ^b^					−0.91	0.74	0.22	−0.07	−0.66	0.76	0.39	−0.05				
Men’s Medication ^b^					0.69	0.85	0.41	0.04	0.02	0.86	0.99	0.00				
Women’s Medication ^b^					−0.21	0.72	0.77	−0.02	0.65	0.71	0.36	0.05				
Men’s Pressure					1.14	1.11	0.30	0.06	−1.59	1.09	0.14	−0.07				
Women’s Pressure					**−3.03**	1.48	0.04	−0.10	**3.08**	1.49	0.04	0.10				
Men’s Current TX ^b^					0.55	0.88	0.53	0.04	−0.44	0.89	0.62	−0.03				
Women’s Current TX ^b^					−1.22	0.72	0.09	−0.10	0.09	0.73	0.90	0.01				
Men’s Avoidant Attachment					**0.39**	0.13	0.00	0.24	**−0.27**	0.13	0.03	−0.16				
Women’s Avoidant Attachment					−0.13	0.12	0.27	−0.08	0.12	0.11	0.31	0.07				
Men’s Anxious Attachment					**0.29**	0.09	0.00	0.20	0.07	0.09	0.41	0.05				
Women’s Anxious Attachment					0.09	0.09	0.32	0.06	**0.59**	0.08	0.00	0.39				
Men’s TX Progress ^a^					−1.31	0.75	0.08	−0.11	0.17	0.74	0.82	0.01				
Women’s TX Progress ^a^					−1.20	0.72	0.10	−0.10	**−1.54**	0.73	0.04	−0.12				
Treatment Length (2–4)					0.33	1.19	0.78	0.02	1.28	1.20	0.29	0.06				

*Note:* TX = treatment. DER = emotion regulation difficulties. RS = relationship satisfaction. S4 = session 4. ^a^ Scores are averaged across treatment. ^b^ Binary coded, see measures for details. All predictors and controls were grand mean centered and measured at session 1 unless specified. All estimates were part of the same model. ***Bolded b***, *p <* 0.05.

**Table 4 behavsci-14-01215-t004:** Estimates, significance rates, and confidence intervals of the indirect effects of Table 3.

	*β*	*p*	*CI*	
Men’s RS → Men’s DER → Men’s Initial Levels of Depressive Symptom Trajectories	0.02	0.67	−0.08	0.13
Men’s RS → Women’s DER → Men’s Initial Levels of Depressive Symptom Trajectories	−0.00	0.67	−0.04	0.01
Women’s RS → Men’s DER → Men’s Initial Levels of Depressive Symptom Trajectories	0.01	0.81	−0.09	0.11
Women’s RS → Women’s DER → Men’s Initial Levels of Depressive Symptom Trajectories	0.00	0.95	−0.02	0.03
Men’s RS → Men’s DER → Men’s Rates of Changes in Depressive Symptom Trajectories	−0.02	0.76	−0.17	0.03
Men’s RS → Women’s DER → Men’s Rates of Changes in Depressive Symptom Trajectories	0.00	0.94	−0.03	0.09
Women’s RS → Men’s DER → Men’s Rates of Changes in Depressive Symptom Trajectories	−0.01	0.86	−0.14	0.05
Women’s RS → Women’s DER → Men’s Rates of Changes in Depressive Symptom Trajectories	0.00	0.99	−0.06	0.05
Men’s RS → Men’s DER → Women’s Initial Levels of Depressive Symptom Trajectories	0.01	0.71	−0.02	0.05
Men’s RS → Women’s DER → Women’s Initial Levels of Depressive Symptom Trajectories	−0.03	0.58	−0.12	0.06
Women’s RS → Men’s DER → Women’s Initial Levels of Depressive Symptom Trajectories	0.00	0.83	−0.03	0.04
Women’s RS → Women’s DER → Women’s Initial Levels of Depressive Symptom Trajectories	0.00	0.93	−0.09	0.09
Men’s RS → Men’s DER → Women’s Rates of Changes in Depressive Symptom Trajectories	−0.01	0.86	−0.17	0.03
Men’s RS → Women’s DER → Women’s Rates of Changes in Depressive Symptom Trajectories	0.02	0.75	−0.01	0.18
Women’s RS → Men’s DER → Women’s Rates of Changes in Depressive Symptom Trajectories	−0.01	0.91	−0.14	0.04
Women’s RS → Women’s DER → Women’s Rates of Changes in Depressive Symptom Trajectories	−0.00	0.96	−0.12	0.05

*Note*: DER = emotion regulation difficulties. RS = relationship satisfaction.

## Data Availability

This is a secondary clinical dataset and is not publicly available (see [20]).

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
