# Peer review of "Emotion Regulation Difficulties as a Mediator Between Relationship Satisfaction Predicting Depressive Symptom Trajectories Among Couples in Couple Therapy"

_behavsci, 2024, doi:10.3390/bs14121215_

Round 1

Reviewer 1 Report

Comments and Suggestions for Authors

1. It is an interesting study, which undoubtedly provides new knowledge on emotional regulation and depressive symptomatology in couples. The fact of studying it in data executed in a clinical context adds value to the conclusions.

2. The development of the empirical framework is quite adequate and clear. The methods are well executed.

3. I especially missed one aspect. Please add a description of the MDT-PRN intervention model and its characteristics, which is very important for such a study. 

4. Please add more references of interest from the last two years.

Author Response

Review 1

COMMENT 1. It is an interesting study, which undoubtedly provides new knowledge on emotional regulation and depressive symptomatology in couples. The fact of studying it in data executed in a clinical context adds value to the conclusions.

RESPONSE 1: We appreciate your comment and also see the use of clinical data as a strength of this study.

COMMENT 2: The development of the empirical framework is quite adequate and clear. The methods are well executed.

RESPONSE 2: Thank you.

COMMENT 3: I especially missed one aspect. Please add a description of the MDT-PRN intervention model and its characteristics, which is very important for such a study. 

RESPONSE 3: The MFT-PRN is a data-collection of clinical data across clinics across the United States and not an intervention model. This is clinical data from naturalistic settings, meaning real-world clinical data. Although therapist interventions are assessed, this type of data is therapist reported. Our communications with the leaders of the MFT-PRN note that these variables are severely under-reported. In essence, it is known that clinical interventions are used due to the setting, but the specific interventions are a limitation of the data. We noted this in the limitations section.

COMMENT 4: Please add more references of interest from the last two years.

RESPONSE 4: Thank you. We have added some references from the last two years.

Reviewer 2 Report

Comments and Suggestions for Authors

This manuscript reports on a study that examined emotional regulation as a mediator between relationship satisfaction and depression trajectories in couple therapy.  I love the fact that the study uses clinical data and also uses dyadic data.  I like the research questions, and I think that the authors did a nice job of laying out the case for their research questions. I found the manuscript to be very well written.  I also believe that the statistical analysis was well designed and appropriately executed.  I only have a few comments and suggestions.

1.       On page 8 (line 303), the authors state that the depressive symptoms decreased “0.17 units every four sessions over the course of couple therapy”.  However, they don’t provide the beta for that relationship, which they do for the other relationships that they report.

2.       On the same page (line 328), the authors report the initial level of depressive symptoms at session one.  I think they meant session eight, because that is what they reported for the men, and that is also what the analysis plan indicated.

3.       On page 9 (lines 48-52), the authors report the results of the mediation model for trajectories of depressive symptoms, but they don’t report them for the initial levels of depressive symptoms at session 8, which is part of the analysis model. 

4.       Just a minor point:  in the reference list, the K references are listed before the J references. 

5.       I was a little confused with the discussion about why there were gender differences in the association between emotional regulations difficulties and the trajectory of depressive symptoms during the course of therapy.  Specifically, higher levels of women’s difficulties with emotional regulations were predictive of their own and their partner’s decreasing trajectory of depressive symptoms.  On the other hand,  there were no significant associations between men’s level of emotional regulation difficulties and trajectories of depressive symptoms. I didn’t think that the two paragraphs that the authors provide to try to explain these findings are very helpful.  I think there are other possible explanations that may be more insightful. 

For example, I think it would be helpful to report the mean levels of men’s and women’s emotional regulation difficulties.  Is there a difference?  And what does other research say about gender differences in emotional regulation difficulties.  I’m wondering because there is research showing that initial more severe levels of relationship are related to greater subsequent improvement in therapy.  Perhaps the same thing might be happening here. Also, does previous research suggest that there are gender differences in the effect of emotional regulation difficulties on depressive symptoms?  That train of thought might be worth pursuing.

Those are just a couple of ideas.  My point is I encourage the authors to do some more digging and exploring to try to come up with some possible reasons for these gender differences.  

Author Response

Review 2

COMMENT 1: This manuscript reports on a study that examined emotional regulation as a mediator between relationship satisfaction and depression trajectories in couple therapy.  I love the fact that the study uses clinical data and also uses dyadic data.  I like the research questions, and I think that the authors did a nice job of laying out the case for their research questions. I found the manuscript to be very well written.  I also believe that the statistical analysis was well designed and appropriately executed.  I only have a few comments and suggestions.

RESPONSE 1: Thank you for comments and review of this manuscript.

 COMMENT 2: On page 8 (line 303), the authors state that the depressive symptoms decreased “0.17 units every four sessions over the course of couple therapy”.  However, they don’t provide the beta for that relationship, which they do for the other relationships that they report.

RESPONSE 2: We revised the sentence to be clear that 0.17 was the beta.

COMMENT 3: On the same page (line 328), the authors report the initial level of depressive symptoms at session one.  I think they meant session eight, because that is what they reported for the men, and that is also what the analysis plan indicated.

RESPONSE 3: Good catch! Yes, that was session eight instead of one. We have revised the sentence to reflect session eight.

COMMENT 4: On page 9 (lines 48-52), the authors report the results of the mediation model for trajectories of depressive symptoms, but they don’t report them for the initial levels of depressive symptoms at session 8, which is part of the analysis model. 

RESPONSE 4: This is helpful to know. We wrote the sentences to include trajectories as a way to briefly convey both the initial and rates of changes of depressive symptoms trajectories. We have revised the sentences to clearly indicate initial levels and rates of changes of depressive symptom trajectories.

COMMENT 5: Just a minor point:  in the reference list, the K references are listed before the J references. 

RESPONSE 5: Thanks for noting this. We have made the revision.  

COMMENT 6: I was a little confused with the discussion about why there were gender differences in the association between emotional regulations difficulties and the trajectory of depressive symptoms during the course of therapy.  Specifically, higher levels of women’s difficulties with emotional regulations were predictive of their own and their partner’s decreasing trajectory of depressive symptoms.  On the other hand,  there were no significant associations between men’s level of emotional regulation difficulties and trajectories of depressive symptoms. I didn’t think that the two paragraphs that the authors provide to try to explain these findings are very helpful.  I think there are other possible explanations that may be more insightful. 

For example, I think it would be helpful to report the mean levels of men’s and women’s emotional regulation difficulties.  Is there a difference?  And what does other research say about gender differences in emotional regulation difficulties.  I’m wondering because there is research showing that initial more severe levels of relationship are related to greater subsequent improvement in therapy.  Perhaps the same thing might be happening here. Also, does previous research suggest that there are gender differences in the effect of emotional regulation difficulties on depressive symptoms?  That train of thought might be worth pursuing.

RESPONSE 6: Thank you for your suggestion. We have added a paragraph describing the possible explanation of gender differences with emotion regulation. The literature on emotion regulation differences is interesting and offered some help. However, our averages between partners were similar. We believe this to be a spring board for future research to explore.

COMMENT 7: Those are just a couple of ideas.  My point is I encourage the authors to do some more digging and exploring to try to come up with some possible reasons for these gender differences.  

RESPONSE 7: Thank you for your encouragement!